# JointNet: Extending Text-to-Image Diffusion for Dense Distribution Modeling

**Jingyang Zhang**[1], **Shiwei Li**[1], **Yuanxun Lu**[3*], **Tian Fang**[1], **David McKinnon**[1], **Yanghai Tsin**[1], **Long Quan**[2†] **Yao Yao**[3‡]

[1]Apple, [2]The Hong Kong University of Science and Technology, [3]Nanjing University

## Abstract

We introduce JointNet, a novel neural network architecture for modeling the joint distribution of images and an additional dense modality (e.g., depth maps). Joint-Net is extended from a pre-trained text-to-image diffusion model, where a copy of the original network is created for the new dense modality branch and is densely connected with the RGB branch. The RGB branch is locked during network fine-tuning, which enables efficient learning of the new modality distribution while maintaining the strong generalization ability of the large-scale pre-trained diffusion model. We demonstrate the effectiveness of JointNet by using RGBD diffusion as an example and through extensive experiments, showcasing its applicability in a variety of applications, including joint RGBD generation, dense depth prediction, depth-conditioned image generation, and coherent tile-based 3D panorama generation.

## 1 Introduction

Recent progress in text-to-image diffusion models has demonstrated extraordinary capabilities in generating visually astonishing images from given text prompts (Ho et al., 2020; Rombach et al., 2022). The technique has shown great potential in defining a new way of visual content creation and has been applied in several vision tasks including image generation, inpainting, editing, and few-shot adaption. On the other hand, except for modeling the data distribution of pure RGB images, there are also great demands in learning the joint distribution of pixel-wise dense labels along with the image. Take depth data as an example (Fig. 1), an RGBD diffusion model is able to unlock a wider range of applications such as joint RGBD generation, dense depth prediction, depth-conditioned image generation, and large-scale tile-based 3D panorama generation.

One simple solution for joint distribution modeling is to train a multi-channel (e.g., 4-channel for RGBD) diffusion model directly from labeled image pairs. However, the number of labeled dense images ($\sim$ 2M depth maps used in MiDaS (Ranftl et al., 2021)) is far less than pure text-image pairs (5B images in LAION (Schuhmann et al., 2022)), making it hard to achieve a joint diffusion model which can be generalized to different kinds of scenes. Alternatively, fine-tuning a pre-trained large-scale text-image diffusion model (e.g., Stable Diffusion (Rombach et al., 2022)) for different tasks has been well-explored in recent years. It is possible to migrate the pre-trained Stable Diffusion model for image-related tasks such as inpainting and editing, by only fine-tuning the model using a relatively small set of data.

In this work, we aim to model the joint image and dense label distribution by extending from a pre-trained text-to-image diffusion model. We propose JointNet, a novel network architecture for efficient joint distribution fine-tuning. A copy of the original diffusion network is created for the dense label branch and is connected with the RGB branch. Inspired by ControlNet (Zhang & Agrawala, 2023), we fix the original RGB branch during JointNet training, while fine-tuning only the dense label branch and connections in between. Such design enables effective joint distribution learning while maintaining the strong generalization ability of the pre-trained text-image diffusion model. We

---

*Working as an intern at Apple

†Working as a consultant at Apple

‡Corresponding author, work done while working at Apple

demonstrate by taking an RGBD diffusion as an example that, a joint distribution modeling model is able to unlock a variety of important applications, including joint data generation, bi-directional dense prediction, and coherent tile-based joint data generation.

To summarize, major contributions of the paper include:

- We introduce a novel approach to model joint distributions of images and dense labels (e.g., depth, normals).
- We present an efficient network architecture that enables the creation of a joint generative model by fine-tuning pre-trained text-to-image diffusion models, preserving their high-quality image generation capabilities.
- We demonstrate the versatility of the joint generative model by showcasing its effectiveness in various downstream tasks, expanding its potential utility in computer vision applications.

## 2    RELATED WORK

### 2.1    DIFFUSION MODELS

Sohl-Dickstein et al. (2015) first proposes to model the distribution of a dataset by iteratively corrupting its structure and then learning to reverse this process. The reverse process can then be used to generate samples from white noise. Following the denoise diffusion process, a variety of works have been proposed in recent years, including effective training strategy (Ho et al., 2020), reduced number of reverse steps (Song et al., 2020), and input compression for high-res generation (Rombach et al., 2022). Compared to variational autoencoders (Kingma & Welling, 2013) that learn both the forward and the reverse process, diffusion models only learn the latter one. Also, compared to generative adversarial networks (GAN) (Goodfellow et al., 2014) that involve a discriminator network, diffusion models use a simpler learning objective. These two improvements both reduce the difficulties of training, making very large-scale training practical.

### 2.2    EXTENDING DIFFUSION MODELS BY FINE-TUNING

The fine-tuning approaches for large-scale generative models can be roughly classified into two categories. The first type of fine-tuning is to introduce a new concept to the already captured data distribution by a small amount of data. The key to this task is to prevent the model from overfitting the new data and forgetting the original knowledge. This issue can be mitigated by limiting the trainable parameters (Ha et al., 2017; Gal et al., 2022; Hu et al., 2021) or introducing regularization (Ruiz et al., 2023). The second type of fine-tuning is to support additional conditioning signals. This typically requires moderate-scale fine-tuning and an extended dataset with the corresponding control signals: Zero-1-to-3 (Liu et al., 2023) introduces camera pose condition for better novel view generation; SDXL (Podell et al., 2023) uses image size condition to enhance the generation of non-squared images; The depth conditioned Stable Diffusion (Rombach et al., 2022) concatenates a depth map to the image to be denoised and extends the number of channels of the first convolution layer with zero initialization; ControlNet (Zhang & Agrawala, 2023) proposes a general extension method by copying the original model to process the conditioning signals. The original parameters are fixed during fine-tuning, thus the training scale can be reduced without destroying the original knowledge. Inspired by ControlNet, we aim to extend a pre-trained diffusion model to jointly generate dense labels by moderate-scale fine-tuning while preserving the generation quality.

### 2.3    ZERO-SHOT ADAPTATION TO DOWNSTREAM TASKS

Recent works have shown that diffusion models can be adapted to tasks other than image generation without fine-tuning. SDEdit (Meng et al., 2021) performs image retouching, including style transfer and detail refinement, by partially corrupting the input images and denoising them back. Repaint (Lugmayr et al., 2022) enables image inpainting by blending partially denoised image with the noise-corrupted input image according to the inpainting mask, in order to preserve the content outside the inpainting mask and ensure the denoising process is conditioned on the unchanged region. Tile-based diffusion (Bar-Tal et al., 2023; Lee et al., 2023; Jiménez, 2023) generates very high-resolution images by cutting the canvas into tiles, applying synchronized denoising, and aggregating the update

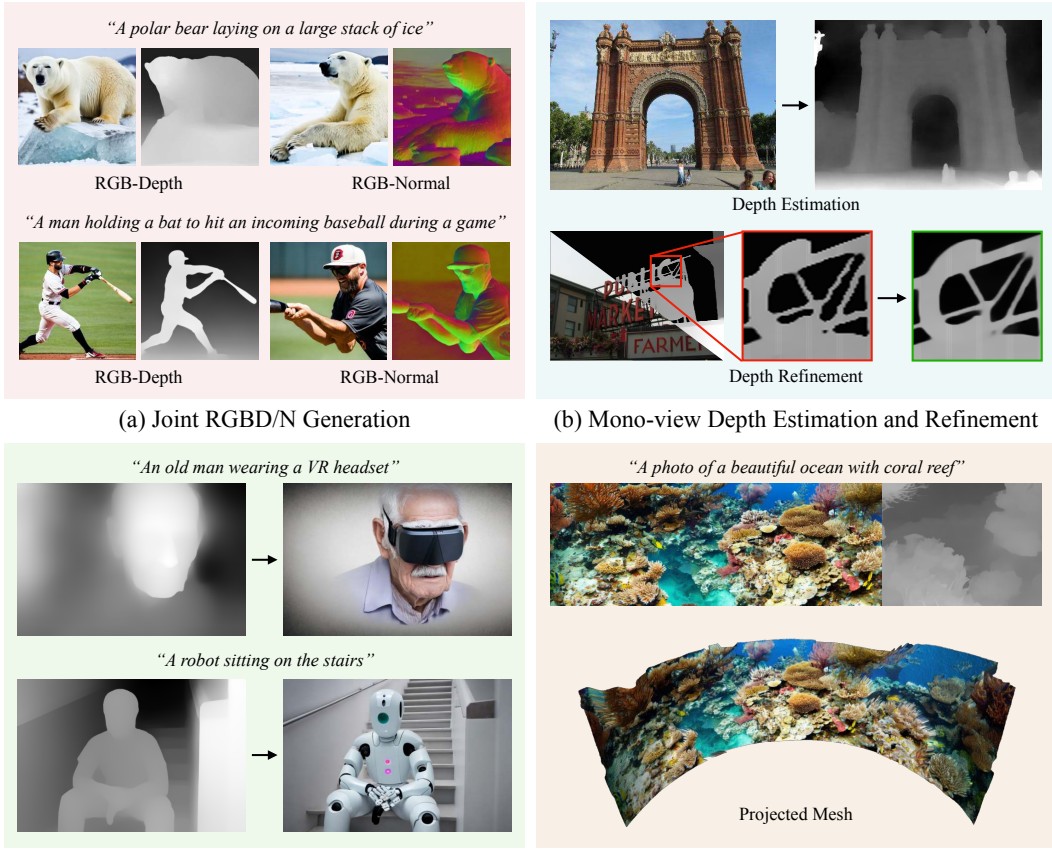

Figure 1: Various tasks achieved by our joint diffusion model.

by averaging. In this paper, we demonstrate that with the proposed JointNet, a joint diffusion model can be applied to various applications related to the new dense modality, and we extensively evaluate our model on these derived tasks.

## 3 METHODOLOGY

In this section, we introduce JointNet, an extension to existing diffusion models that enable the joint generation of RGB images and an additional dense modality. In Sec. 3.1, we present a detailed analysis of the proposed JointNet architecture, which is designed to achieve high-quality joint generation without sacrificing the performance in original RGB domains. Specifically, we discuss the latent diffusion case, including the conversion between image space and latent space in Sec. 3.2. The data preparation process is discussed in Sec. 3.3. Lastly, we provide training details in Sec. 3.4.

### 3.1 JOINTNET

As reviewed in Sec. 2.2, there are many mature solutions to fine-tune existing diffusion models for including out-of-distribution samples or accepting additional conditions. The former type does not introduce additional trainable weights, and the latter type often uses zero-initialized convolution layers to transform the input condition before joining the main data flow of diffusion model (Rombach et al., 2021; Zhang & Agrawala, 2023). These two types of fine-tuning cases share a common fact: the output of the extended model is not altered from the original one in the first iteration of training. This insight, termed as *output preserving principle* in the following sections, is the key guideline of our design of JointNet.

A trivial extension method that barely fulfills the requirement of input and output shape is to increase the number of channels of the first and the last layer (Fig. 2(a)). The extra trainable parameters in the input layer can be initialized to zero as in the case of model extension for additional conditioning.

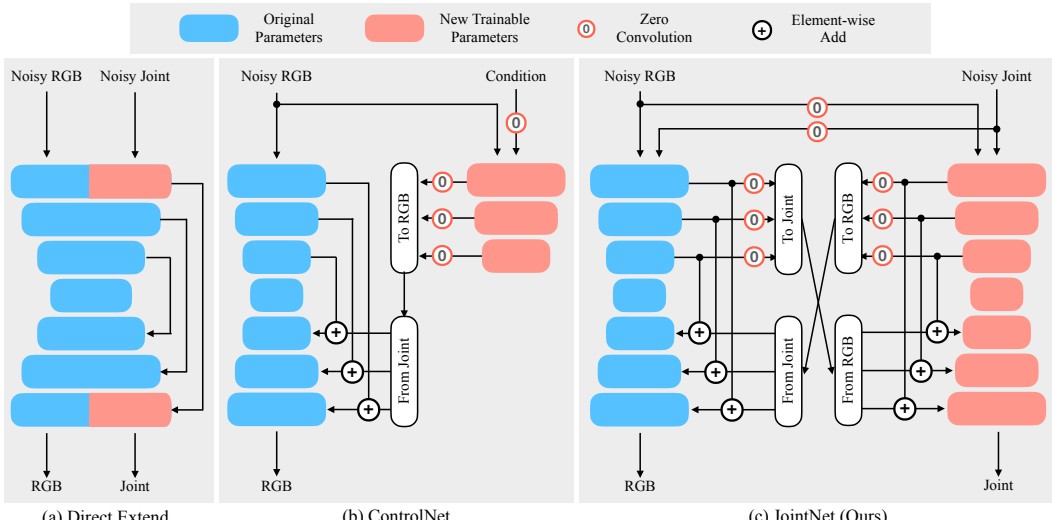

Figure 2: Comparison of network architectures. (a) A trivial solution of directly extending the number of channels in input and output layers. (b) The structure of ControlNet (Zhang & Agrawala, 2023) from which our design is inspired. (c) The proposed JointNet structure.

For the output layer, the parameters can be initialized by zeros, random numbers, or a copy of the original ones. However, none of these last layer initializations achieve the principle mentioned above: The first two choices output zero or random estimation, and the last one predicts the noise for the RGB input instead of the joint modality. During training, the large loss of the joint output will make the main body of the network dramatically deviate from the original state, which leads to catastrophic forgetting. We have observed in experiments that the model suddenly loses the ability to generate reasonable images in early steps and then gradually recovers by learning from the new dataset from scratch. Examples can be found in Sec. A.1.

Inspired by ControlNet (Zhang & Agrawala, 2023), we use a copy of the original diffusion model. As shown in Fig. 2, each branch takes a noisy map in the corresponding modality as input, and estimates the noise in it. Text prompts and time conditioning are still fed into the cross-attention modules (which are omitted in the figure). To establish information communication, the intermediate results in the downsampling and middle stages, which are originally used as residual inputs in the upsampling stage, are also sent to the other branch. In the meantime, the current branch will receive the intermediate results from the opposite side. Such information will be added to the intermediate results from the current branch and sent to the upsampling stage together. In addition, the input of the other branch is also fed into the current one at the beginning. All these exchanging tensors are transformed by an additional zero-initialized convolution layer.

We validate that the proposed design fulfills the output preserving principle. In the first iteration of training, the zero-initialized layers vanish the exchanging information, and thus both branches estimate the noise independently using the original parameters. For common modalities like depth or normal, the pre-trained RGB model actually has a limited ability of denoising them because the dataset for training the model (e.g. LAION-5B (Schuhmann et al., 2022)) contains a small number of these data. In summary, neither of the branches will experience a sudden rise of loss, and thus the model can adapt smoothly to the new objective.

Note that the proposed extension method is independent of diffusion architectures. In this paper, we conducted experiments both on the Stable Diffusion (Rombach et al., 2022) and the Deepfloyd-IF (DeepFloyd, 2023).

## 3.2 LATENT DIFFUSION MODELS

For latent diffusion models like the Stable Diffusion (Rombach et al., 2021), the diffusion process is performed in latent space defined by a variational autoencoder. To extend such models, we also need to encode the joint input into a latent space. Previous method (Stan et al., 2023) fine-tunes the autoencoder to support additional input channels. However, changing the autoencoder will lead to a

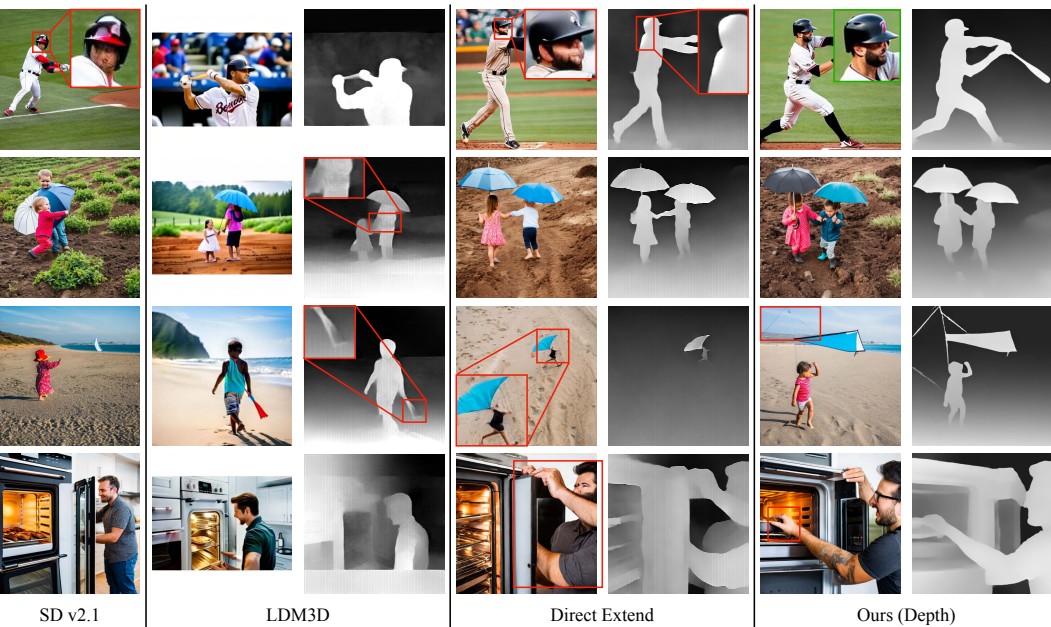

SD v2.1          LDM3D          Direct Extend          Ours (Depth)

Figure 3: Qualitative comparison of jointly generated samples. Our model successfully preserves the RGB generation quality of the base model and generates reasonable joint modality.

mismatch between the new latent space and the one already captured by pretrained diffusion models. Therefore, we simply use the autoencoder for RGB to encode and decode the joint modalities. This may result in a performance drop. Potential solutions include fine-tuning the decoder part of the autoencoder system for joint modalities.

## 3.3 DATA PREPARATION

We perform training on the COYO-700M dataset (Byeon et al., 2022) containing image-caption pairs as well as various metadata including properties like image size and derived evaluations such as CLIP (Radford et al., 2021) similarity, watermark score and aesthetic score (Schuhmann, 2022). We select samples with both size greater than 512, watermark score lower than 0.5, and aesthetic score greater than 5. The number of the selected samples is around 65M, but the training process will not consume the whole filtered dataset within the designated time.

The joint modalities are estimated online. In this paper, we test depth and normal. We use MiDaS v2 (Ranftl et al., 2021) and Omnidata (Eftekhar et al., 2021; Kar et al., 2022) for depth and normal estimation respectively. Both models are based on DPT-hybrid (Ranftl et al., 2021) architecture. The RGB images will first be resized to 384x384 before fed into the network, and the estimated depth or normal maps are resized to target training size.

## 3.4 TRAINING

The training loss is similar to previous works (Rombach et al., 2022; Ho et al., 2020; Song et al., 2020): the neural network is expected to predict the noise used to corrupt the data, and loss is given by the L2 difference between the actual noise and the predicted one. For JointNet, we denoise both image $x$ and another joint dense map $y$ with the original network $\epsilon_\theta$ and its copy $\epsilon_\phi$ respectively, and the total loss is the sum of the losses from these two branches. Let $x_t, y_t$ be the data corrupted by $\epsilon_x$ and $\epsilon_y$ at time $t$, and $c$ represent any conditioning signals, the loss is given by

$$L = \mathbb{E}_{(x,y,c);\epsilon_x,\epsilon_y \sim N(0,1);t}[\|\epsilon_x - \epsilon_\theta(x_t, y_t, c, t)\|_2^2 + \|\epsilon_y - \epsilon_\phi(y_t, x_t, c, t)\|_2^2]. \quad (1)$$

We train the network on 64 NVidia A100 80G GPUs for around 24 hours. The sample resolution is 512x512 and the batch size is 4 on each GPU. The model is trained with a learning rate of 1e-4 for 10000 steps with 1000 warmup steps. We adopt a probability of 15% to drop the text conditioning (Ho & Salimans, 2022) and apply noise offset (Lin et al., 2023) of 0.05. The original parameters in the RGB branch are frozen.

Optionally, we can further fine-tune the model including the originally frozen parameters with a learning rate of 1e-5. Similar to ControlNet (Zhang & Agrawala, 2023), the text drop rate is raised to 50% in order to let the model learn mutual conditioning between RGB and joint modalities. This fine-tuned model is set as our default setting in the following experiments.

# 4 APPLICATIONS

In this section, we introduce downstream tasks enabled by the examplar joint RGBD diffusion model. In Sec. 4.1, we show that the bidirectional dense prediction, including mono-view depth estimation and depth-conditioned image generation, can be achieved by the channel-wise inpainting of the joint RGBD model. In Sec. 4.2, we propose a tile-based generation technique designed to generate images with larger sizes and varying aspect ratios, which may be suboptimal for the model to generate directly. Specifically, we showcase in Sec. 5.3 the capability of generating RGBD panorama as a 3D immersive environment.

## 4.1 BI-DIRECTIONAL DENSE PREDICTION VIA CHANNEL-WISE INPAINTING

One of the most significant applications derived from text-to-image diffusion is image inpainting. In a common inpainting task, an input image and a mask are provided, and the model modifies the content within the mask based on text prompts while preserving the region outside the mask unchanged. Algorithmically, this can be achieved using the original model with a modified generation pipeline (Lugmayr et al., 2022). At each step, the partially denoised result is blended with the noise-corrupted input according to the mask, which ensures that the content outside the inpainting area remains unchanged and the inpainting process is conditioned on the original content. To enhance the diffusion model's ability for inpainting tasks, we can further fine-tune the model with masked input as additional conditioning (Rombach et al., 2022). The strategies can be directly generalized to our RGBD diffusion model. For RGBD image editing, we just set the same mask for image and depth, and apply the inpainting pipeline. Examples can be found in Sec. A.3.

Except for symmetric masks for both image and depth, what is interesting about the joint diffusion model is its versatility in employing asymmetric masks for image and the dense modality, which opens up additional possibilities for various applications. Given a joint RGBD generation model, the task of depth-conditioned image generation and mono-view depth estimation can both be modeled as channel-wise inpainting. The former can be achieved by setting pure white for image mask and pure black for depth mask, and reverse the two for the latter task.

In practice, we found that the original model cannot perform well in channel-wise inpainting. So we further fine-tune the model with masked input as additional conditioning. We resume from the checkpoint of the first stage mentioned in Sec. 3.4 and adopt the same strategy as the second stage except for the extra input. Note that the fine-tuned model is still capable of generating RGBD images by setting pure white for both masks.

## 4.2 TILE-BASED GENERATION WITH CONTENT COHERENCE

Given that the diffusion models are only trained by squared samples with size 512, it will be suboptimal to directly generate image with resolution and aspect ratio dramatically deviated from the training setting. Alternatively, tile-based generation strategy (Bar-Tal et al., 2023; Jiménez, 2023) can mitigate this problem, with the cost of reduced consistency across the tiles. Perceptual loss (Zhang et al., 2018; Lee et al., 2023) can be applied between the tiles to encourage the consistency.

In this work, we use a different strategy: we combine both whole-image and tile-based denoising. In the first 40% of the whole denoising process, we add the whole image as another tile, which also participates in the update aggregation of tiles. The weight of this whole-image tile is increased to 5. The motivation is to first determine a rough layout of the image, instead of letting the tiles be denoised independently and get deviated from the beginning. In the rest of the denoising steps, this whole-image tile is removed to avoid its negative influence on the details. Other tricks include applying random offset to all the tiles in each step so as to reduce the overlap between neighboring tiles, and decaying the aggregating weight linearly to 0 at the tile boundaries for smooth transition.

|         | SD v2.1 | LDM3D | Direct Extend | Ours (Depth) | Ours (Normal) |
|---------|---------|-------|---------------|--------------|---------------|
| FID↓    | 16.15   | 32.13 | 14.15         | 14.38        | 15.62         |
| IS↑     | 39.85   | 29.40 | 37.08         | 38.33        | 39.71         |
| CLIP↑   | 26.17   | 26.54 | 25.74         | 25.71        | 25.87         |

Table 1: Quantitative comparison of jointly generated samples. Our model has comparable performance compared with the base model.

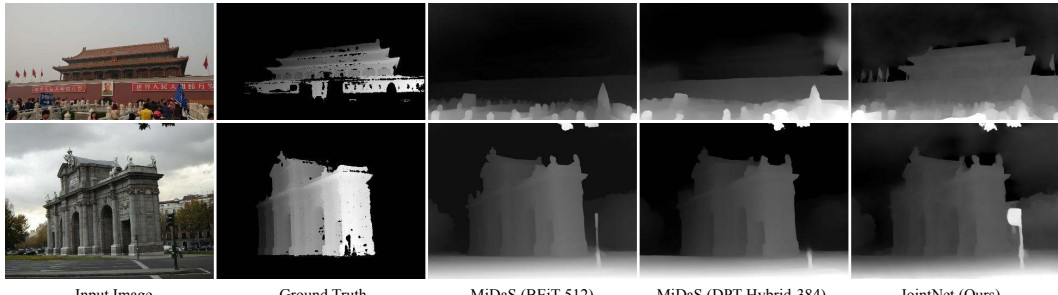

Input Image       Ground Truth       MiDaS (BEiT-512)       MiDaS (DPT-Hybrid-384)       JointNet (Ours)

Figure 4: Qualitative comparison of mono-view depth estimation on the MegaDepth dataset (Li & Snavely, 2018).

## 5 EXPERIMENTS

### 5.1 JOINT GENERATION

The basic application of the model is to jointly generate image and its depth/normal. We compare the results with three other methods. The first one is Stable Diffusion v2.1 from which our model is fine-tuned, which can serve as a reference of RGB quality. The rest two methods, LDM3D (Stan et al., 2023) and the baseline introduced in Fig. 2(a) (named as *Direct Extend* in this section), both directly extend the original model for the additional generation channels.

Fig. 3 shows the qualitative results. The prompts used to generate these examples are all from the MSCOCO validation set. We use the same seed 7 for all the methods. For RGB quality, the results from LDM3D may have white margin and blurred content similar to interpolated low-res images. This may be related to the data filtering and preprocessing strategy. The *Direct Extend* model does not perform well for details like faces, limbs and other small structures. The proposed method also produce wrong details occasionally, but the quality is comparable with the base model. For depth quality, the results from LDM3D may have wrong values and quantization artifacts. And the *Direct Extend* model may have overshot values at object boundaries. More results from JointNet with other base models and modalities can be found in Sec. A.4.

Quantitatively, we compare the Fréchet inception distance (FID) (Heusel et al., 2017), inception score (IS) (Salimans et al., 2016) and CLIP similarity (Radford et al., 2021) of the generated RGB over a prompt collection of size 30K sampled from the MSCOCO (Lin et al., 2014) validation set. The results are listed in Tab. 1. All the models achieves comparable scores except LDM3D which have larger FID and lower IS. The low IS score has been discussed in the original paper, and the high FID may be resulted from the white margins. LDM3D achieves the best CLIP similarity score possibly because the model is trained on LAION-400M (Schuhmann et al., 2021) which is filtered by the same metric. The proposed method cannot outperform *Direct Extend* in terms of FID despite of qualitative improvement. One possible reason is that the global image descriptor used by FID cannot capture the difference of the details.

In summary, the proposed fine-tuning method successfully preserves the RGB generation quality of the base model which can only be achieved by very large-scale training.

### 5.2 BIDIRECTIONAL DENSE PREDICTION

In this section, we extensively evaluate the bidirectional image conversion ability of the proposed method. Apart from qualitative examples, we quantitatively evaluate the image-to-depth case by the metrics used in previous mono-view depth estimation systems. Because the training data is prepared

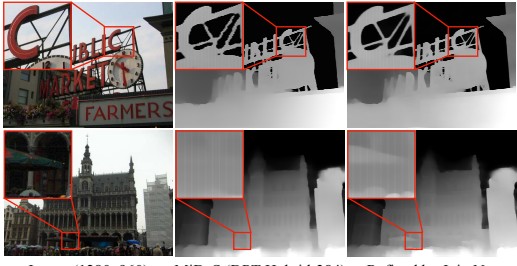

Figure 5: Qualitative results of depth refinement by JointNet. The refined results have sharper edges without aliasing and more distinguishable small objects.

| | AbsRel | RMSE |
|---|---|---|
| MiDaS (BEiT-512) | 0.0903 | **0.0415** |
| MiDaS (DPT-Hybrid-384) | 0.0568 | 0.0601 |
| JointNet (Ours) | 0.1203 | 0.0754 |
| MiDaS + JointNet | **0.0561** | 0.0634 |

Table 2: Quantitative comparison of mono-view depth estimation. The proposed JointNet has comparable performance in terms of RMSE, and the results refined by JointNet achieve the best AbsRel.

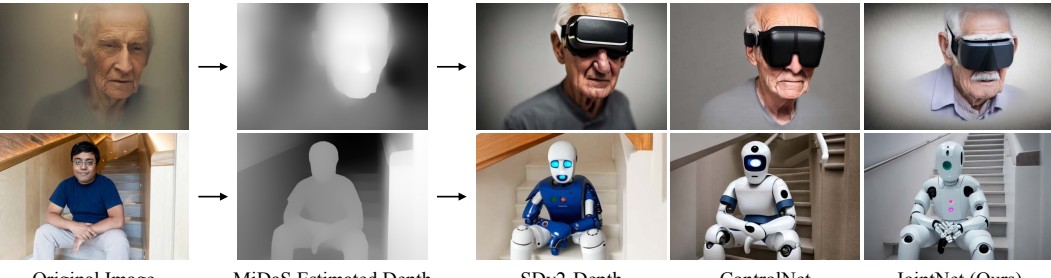

Figure 6: Qualitative results of depth conditioned image generation.

by the state-of-the-art mono-view depth estimation model, we only expect the quality of diffusion depths to be comparable.

**Image-to-depth.** The evaluation of mono-view depth estimation is conducted on the MegaDepth (Li & Snavely, 2018) dataset, which provides filtered multi-view stereo (MVS) depth maps as ground truth. Following MiDaS (Ranftl et al., 2021) and LDM3D(Stan et al., 2023), we first align estimations and ground truths because estimated depths are in disparity space up to scale and shift. For each pair, we take samples from valid pixels (depth values from both maps are greater than 0) and solve the scale and the shift by the least square method. The parameter solving is performed multiple times and we take the median from the results as the final alignment. Then, we evaluate both absolute relative error (AbsRel) in depth space and root mean squared error (RMSE) in disparity space. We compare our method with MiDaS (Ranftl et al., 2021) with two types of backbones, among which the DPT-Hybrid version is used to generate our training data. Quantitative results are listed in Tab. 2 and qualitative results are shown in Fig. 4. Our method is comparable with the two baselines in terms of RMSE, but is worse in terms of AbsRel. Apart from algorithmic aspects, one possible reason is that MiDaS is trained on MegaDepth dataset while our model does not.

Although the depth directly from JointNet is not guaranteed to have very high accuracy, we can take advantage of the iterative nature of diffusion models to refine the depth from MiDaS. This is useful when high-resolution depth is desirable, provided that MiDaS is designed for images with sizes of 512 or 384. To test the depth refinement, we first estimate depth for a given image by MiDaS (DPT-Hybrid-384) at a resolution of 384 and upsample it back to the original size without interpolation (nearest mode). The upsampled depth map is then fed into JointNet and refined by the strategy proposed in SDEdit (Meng et al., 2021). We use denoising strength as 0.4 in the experiments. Because JointNet is also trained with 512 images only, we adopt the tile-based strategy introduced in Sec. 4.2 without the whole-image tile. Qualitative results and comparison with the depth map before the refinement is demonstrated in Fig. 5. Major improvements include sharper edges without aliasing and more distinguishable small objects that are hard to detect at low resolution. The quantitative improvement is not significant as shown in Tab. 2. One possible reason is that edges are filtered out in the ground truth and thus not counted in the statistics.

**Depth-to-image.** For depth-conditioned image generation, we test the examples used by depth-conditioned Stable Diffusion v2 (Rombach et al., 2022) and ControlNet (Zhang & Agrawala, 2023),

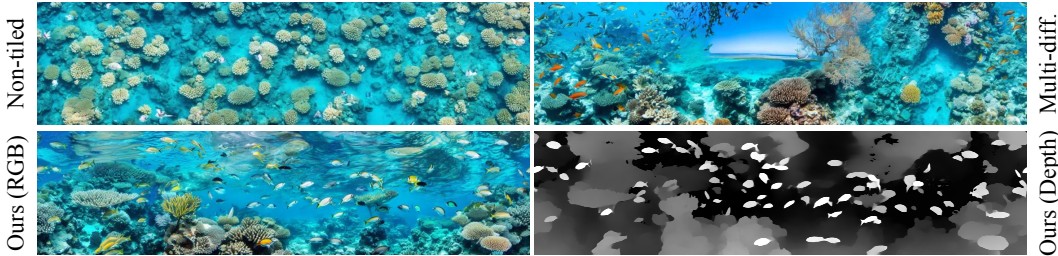

*A photo of a beautiful ocean with coral reef*

Figure 7: Qualitative comparison of generated panoramas. JointNet can produce RGBD panoramas with coherent and meaningful content.

| | Setting | Tile Stride | Time (s) | Intra-LPIPS↓ |
|---|---|---|---|---|
| Non-tiled | Directly generate the whole panorama | - | 12 | 0.507 |
| Multi-Diffusion | Tile-based diffusion | 8 | 49 | 0.641 |
| Sync-Diffusion | Intra-LPIPS loss with weight 20 and decay 0.95 | 16 | 101 | 0.626 |
| Ours | Random offset, full image denoising, boundary decay | 32 | **17** | **0.578** |

Table 3: Quantitative comparison of running time and (RGB) content coherence of panorama generation over 320 samples. JointNet achieves the best content coherence while keeping the lowest time consumption.

and compare the results with these two methods. Qualitative results are illustrated in Fig. 6. The multi-functional JointNet can offer comparable performance for this task.

### 5.3 3D PANORAMA GENERATION

In this task, we generate RGBD panoramas from input text by the tile-based strategy introduced in Sec. 4.2 and convert them into a 3D immersive environment. The target panoramas have a cylindrical projection, span $180°$ horizontally, and have a resolution of 2048x512.

Fig. 1 shows the generated RGBD panoramas and the projected meshes. To verify the content coherence strategies introduced in Sec. 4.2, we evaluate the intra-LPIPS loss (Lee et al., 2023) over 320 samples generated from 16 prompts. For each panorama, we calculate pair-wise perceptual loss for all 6 combinations and average them. Quantitative results are listed in Tab. 3. Among the tile-based methods, our method achieves the lowest intra-LPIPS loss while keeping low time consumption. Qualitative results are illustrated in Fig. 7. The results of *Non-tiled* diffusion contain repetitive contents, while the tile-based methods create meaningful panoramas.

### 6 LIMITATIONS AND FUTURE WORK

**Doubled time consumption.** As JointNet maintains two branches of the diffusion network, the inference time is naturally doubled as well. Possible solutions include distillating the full model copy to a lightweight one.

**More number of modalities.** In this work we explicitly use a copy of the original model to support the second modality. To further support other modalities simultaneously, the default solution is introducing more copy. However, such direct extension would result in even more time consumption. A rigorous investigation for a more elegant solution is needed when extending to multi-modalities.

### 7 CONCLUSION

We have presented JointNet, a novel extension method for fine-tuning existing text-to-image diffusion models for joint dense distribution modeling. A trainable copy of the original network is introduced for the additional modality while the original RGB branch is fixed, which maintains the strong generalization ability from the base model. The joint diffusion model also unlocks various of downstream tasks including bidirectional dense prediction and coherent tile-based joint data generation. The generation ability and the downstream tasks have been extensively evaluated both qualitatively and quantitatively, demonstrating the strong performance and versatility of the proposed system.

## 8 ACKNOWLEDGMENTS

This work is supported by Hong Kong RGC GRF 16206722.

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

## A   APPENDIX

### A.1   VISUALIZATION OF TRAINING PROGRESSION

During training we periodically generate validation samples by the same prompt and seed. Results are shown in Fig. 8. As discussed in Sec. 3.1, the *Direct Extend* method loses the RGB generation capability in early steps, while JointNet successfully preserves it. At the 5000-th step, JointNet is able to generate better depth map than *Direct Extend*.

Moreover, we additionally test the setup where the input is processed by ControlNet while the output is directly extended. The training progression is shown as *ControlNet + Direct Extend*. The RGB generation capability is degraded similarly to *Direct Extend*, showing that the key is the way of handling the output.

### A.2   ADDITIONAL EXPERIMENTS

#### A.2.1   USER STUDY ON JOINT GENERATION QUALITY

We conduct user study on overall quality of generated RGBD images. We compare JointNet with LDM3D, *Direct Extend*, and *ControlNet + Direct Extend* introduced in Sec. A.1. We prepare 100 text prompts sampled from COCO validation set and let user choose the overall best generated RGBD image in terms of image-text consistency, image-depth consistency and the quality of image and depth themselves. The results are shown in Tab. 4. Among all the surveyed methods, JointNet achieves the highest satisfaction rate. We notice that LDM3D achieves the second highest score, which contradicts the qualitative and quantitative conclusion in the paper. After interviewing the participants, the main reason is that text-image consistency plays an important role in their decision making, which is LDM3D's strength, while the blurred image and the artifacts in the depth maps are not significant if the participants do not zoom in the samples.

#### A.2.2   COMPARISON WITH CONTROLNET

In the paper we compare our method with ControlNet and also the depth conditioned Stable Diffusion v2.0 on the depth-to-RGB generation task qualitatively. To make the comparison more comprehensive, we additionally conduct quantitative comparison: we sample 30000 image-text pairs from the COCO validation dataset, estimate depth of them by MiDaS, perform depth-to-RGB generation and compare the FID between the ground truth and the generated images. The results are shown in Tab. 5. The depth conditioned SD v2.0 (*SD v2.0d*) achieves the best performance among all, but the finetuning scale is much larger than other two methods. Our method shows comparable result with ControlNet, which is coincide with the qualitative observation in the paper. The marginal quantitative gap is negligible because of the randomness of generation.

### A.3   IMAGE EDITING

Fig. 9 illustrates examples of joint RGBD image editing by inpainting.

### A.4   ADDITIONAL QUALITATIVE RESULTS

We also train JointNet for depth or normal based on DeepFloyd-IF or Stable Diffusion. Fig. 10 shows generated samples from these models.

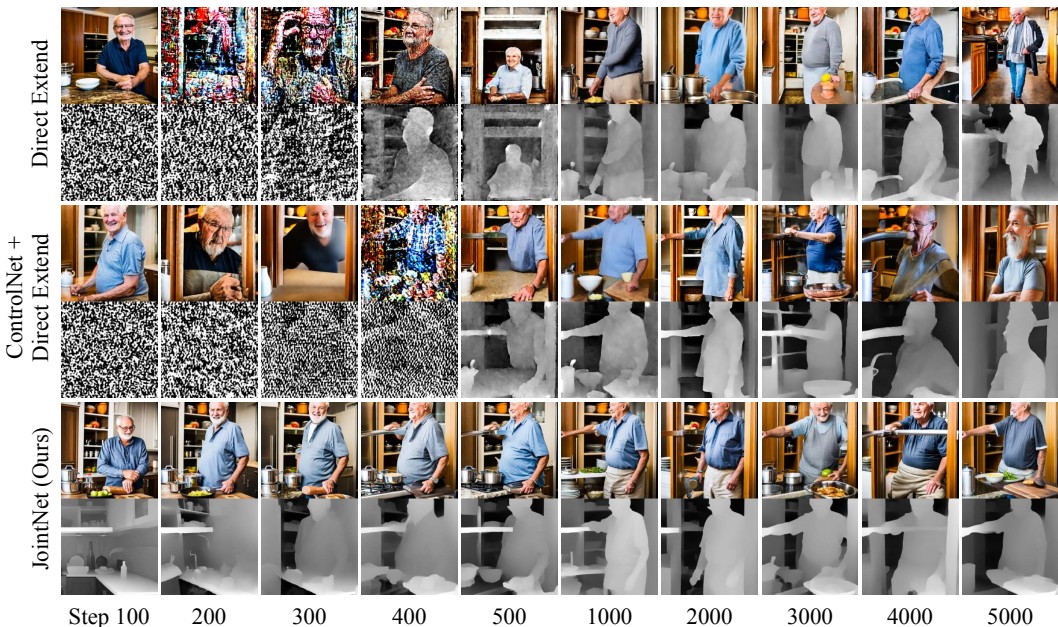

Figure 8: Progression of validation samples. The *Direct Extend* and the *ControlNet + Direct Extend* method lose the RGB generation capability in early steps, while JointNet successfully preserves it.

| LDM3D | Direct Extend | ControlNet + Direct Extend | JointNet (Ours) |
|---|---|---|---|
| 27.0% | 23.7% | 20.2% | **29.1%** |

Table 4: User study of joint generation quality. For each of the 100 sampled text prompts, participants are told to select the best overall generated RGBD sample from 4 choices. Among all the surveyed methods, JointNet achieves the highest satisfaction rate. LDM3D also achieves good result due to its strong text-image consistency.

Fig. 11 shows more results for mono-view depth estimation.

Fig. 12 shows more results for the tile-based panorama generation.

## A.5 PANORAMA REFINEMENT

The panoramas can be further upsampled to a higher resolution. Similar to the depth refinement demonstrated in Sec. 5.2, we first interpolate the RGBD panoramas to the target resolution, and refine both RGB and D by JointNet in the tile-based manner. Qualitative results are shown in Fig. 13. After the refinement, the image have better details, and the depth is sharper and better aligned with the image.

|       | SD v2.0d | ControlNet | JointNet (Ours) |
|-------|----------|------------|-----------------|
| FID   | 8.75     | 10.14      | 10.61           |

Table 5: Quantitative comparison of depth-to-RGB generation. SD v2.0d achieves the best score among all, with the cost of much heavier training. Our method is comparable with ControlNet.

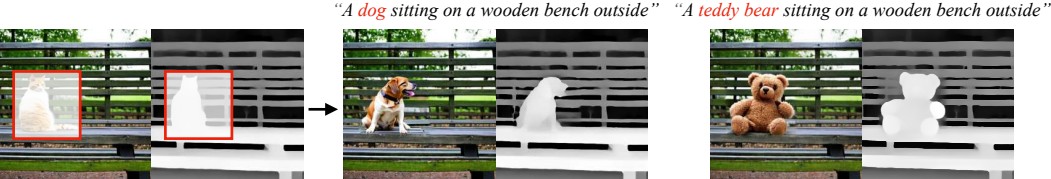

Figure 9: Qualitative examples of joint RGBD editing by inpainting.

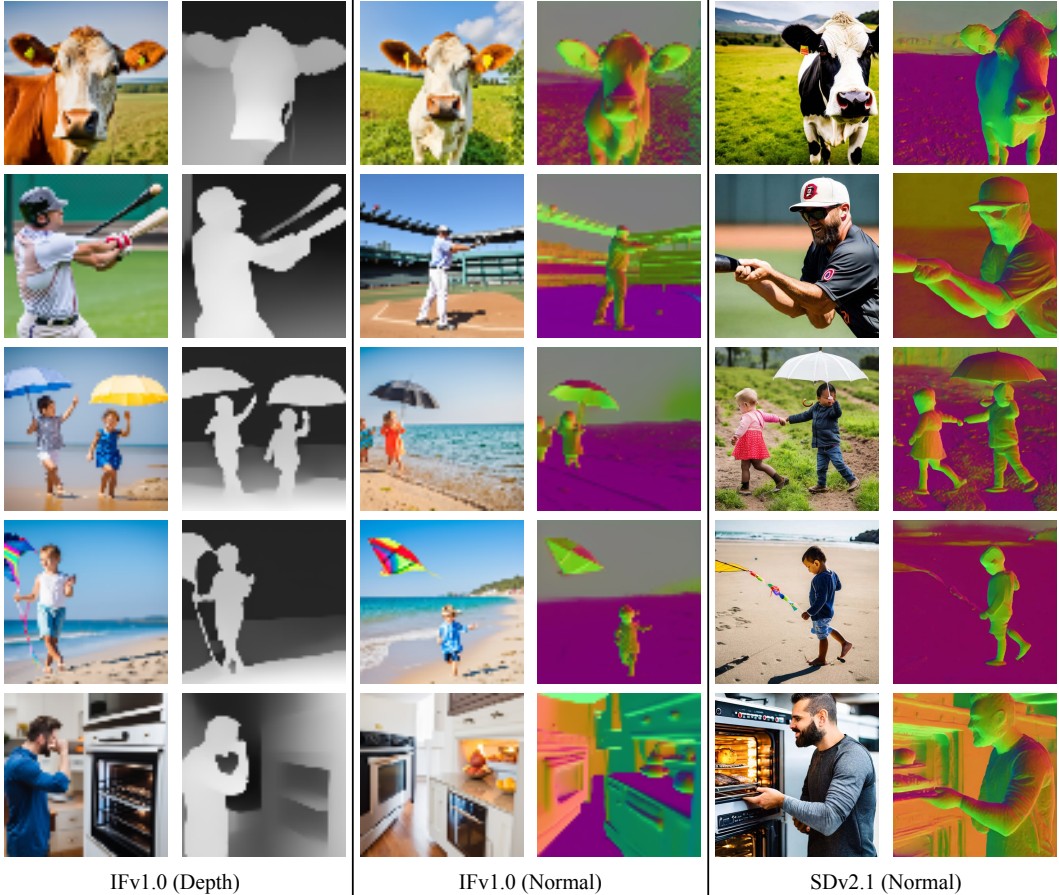

Figure 10: Additional qualitative results of joint depth or normal generation. *IFv1.0* and *SDv2.1* stands for the JointNet fine-tuned from DeepFloyd-IF v1.0 and Stable Diffusion v2.1 respectively.

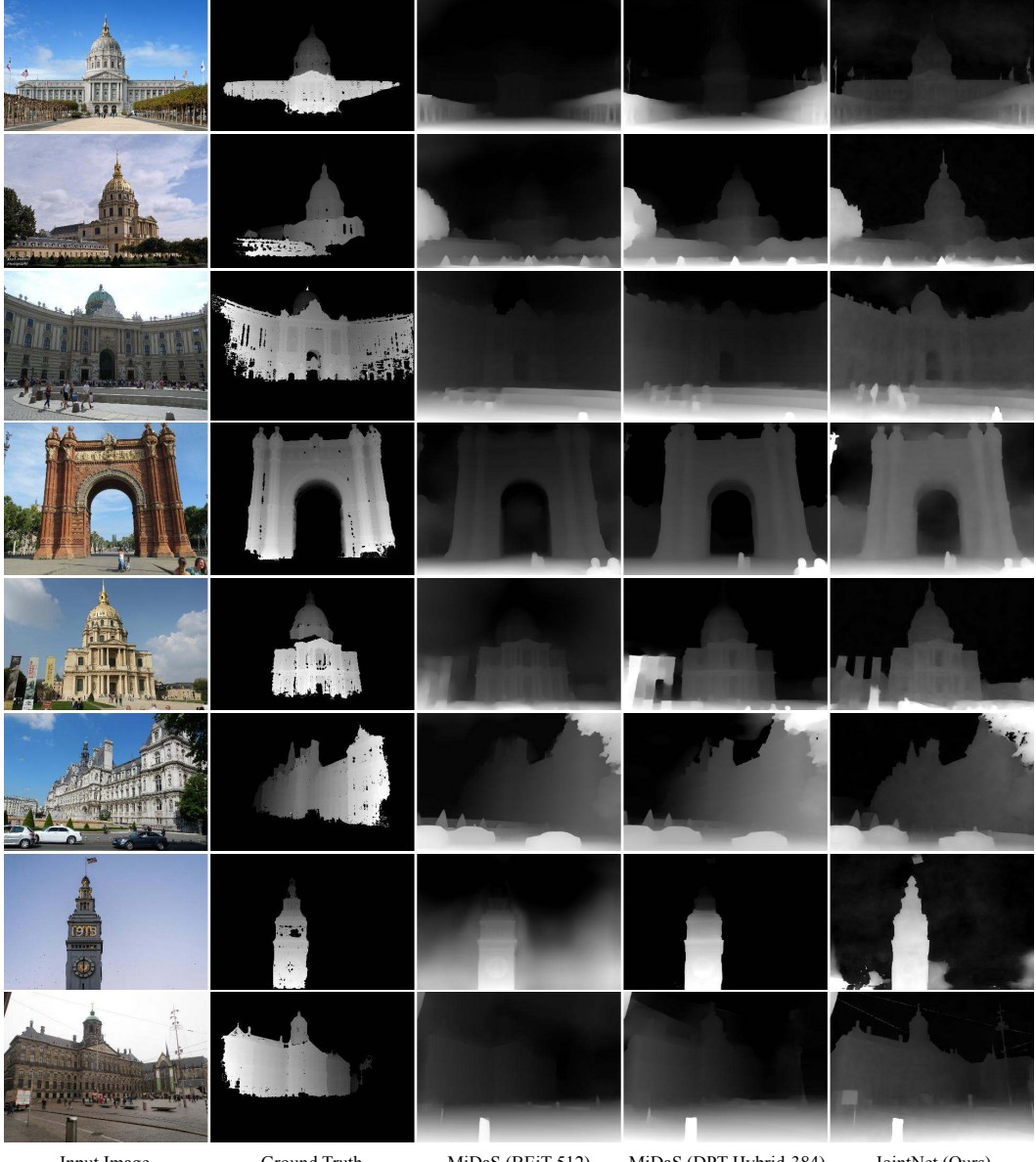

| Input Image | Ground Truth | MiDaS (BEiT-512) | MiDaS (DPT-Hybrid-384) | JointNet (Ours) |

Figure 11: Additional qualitative results of mono-view depth estimation.

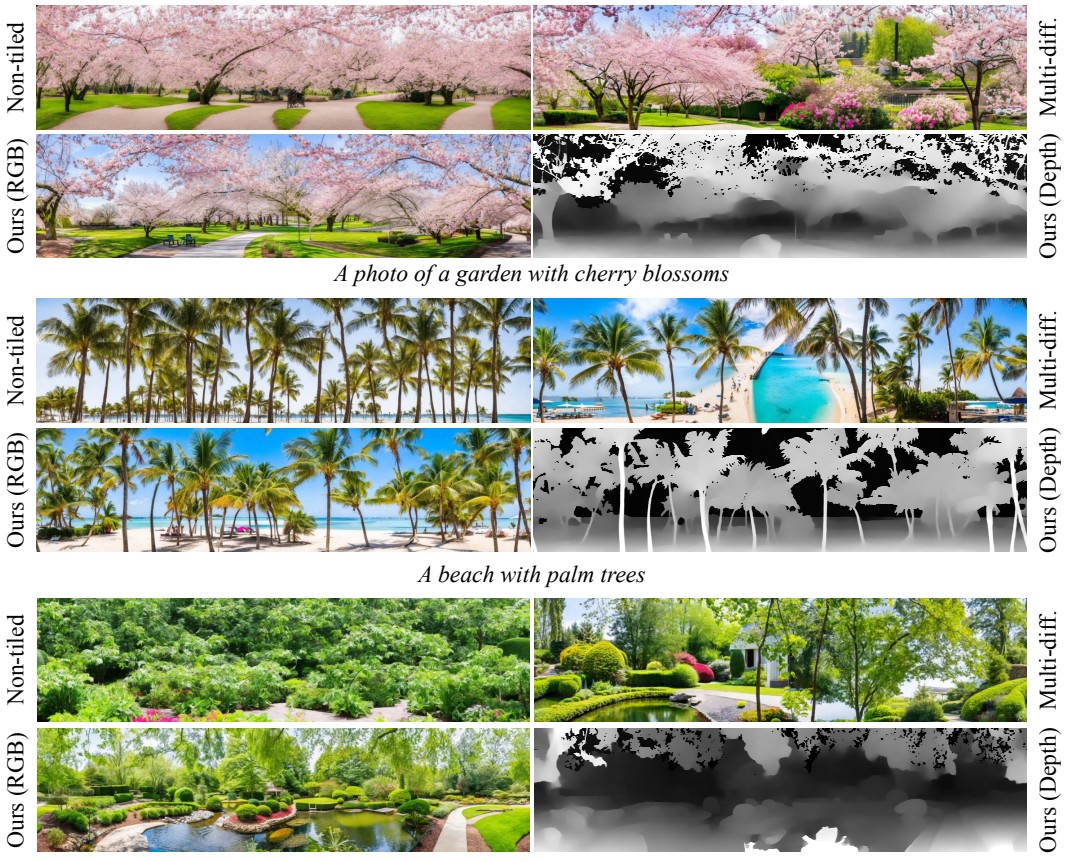

Figure 12: Additional qualitative results of RGBD panorama generation.

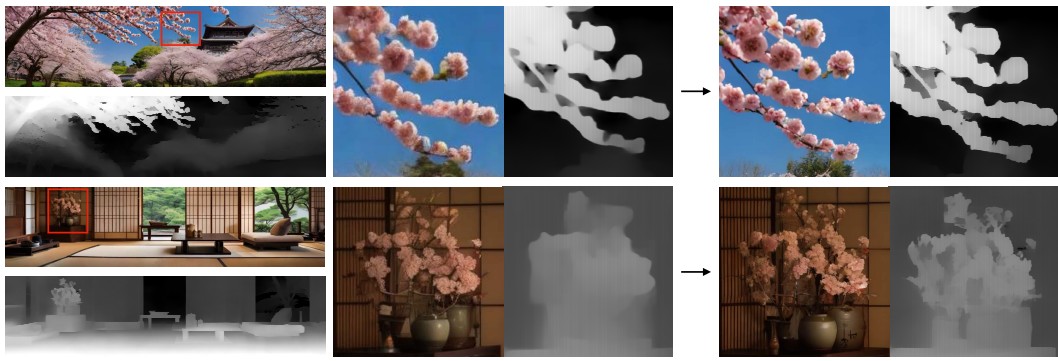

Initial RGBD Panorama                    Refined by JointNet

Figure 13: JointNet can refine details for lowres RGBD panoramas in a tile-based manner.

