# REBUTTAL

## 1 REVIEWER YGUJ

We thank the reviewer for the acknowledgement of the motivation of this work and the importance of the task. Here we address the questions raised by the review as follows.

### 1.1 CLARIFICATION ON THE CONTRIBUTION

We would like to clarify that the core idea is using joint distribution modeling to construct a large vision model capable of various of downstream tasks, instead of pushing the SOTA of each single downstream task. Because training large models is more and more expensive, it is desirable to design a general training objective for a multi-tasking model. For the performance gap of image-to-depth task, because our depth training data is prepared by the mentioned SOTA depth estimation network and we do not directly train on (the training split of) the depth datasets the benchmark uses, our model may not be able to outperform the SOTA depth estimator, which is indeed a limitation. Possible improvements include adding the depth datasets for training and finetuning the VAE for the joint modalities. More explanation can be found in the answer to reviewer *4SM3* (Sec. 4.2).

On the other hand, the joint diffusion model unlocks several new usage such as tile-based consistency handling for high-res RGBD generation (e.g., RGBD panorama), which cannot be achieved by the traditional discriminative model of monocular depth estimation. To summarize, we believe our work demonstrates the potential of the joint training scheme, and there are possible ways of improvement to further push the performance on each downstream task.

## 2 REVIEWER UNZU

We thank the reviewer for the appreciation of the contribution and clear paper writing. Here we address the questions raised by the review as follows.

### 2.1 ADDITIONAL COMPUTATIONAL COMPLEXITY FOR JOINT MODALITIES

Thanks for the suggestion on the possible future improvement and we agree with it. Because the additional branch needs to be trained to adapt to a new domain after all, preserving the original knowledge is less important than the RGB branch, and thus we can apply the direct extend strategy to merge multiple modalities into the joint branch. One disadvantage is the reduced flexibility. If each modality has its own JointNet, users can pick only the needed ones and use them in a plug-and-play manner just like multi-ControlNet.

## 3 REVIEWER CBX7

We thank the reviewer for the acknowledgement of the novelty of the network and finetuning design, and the capability of handling various of downstream tasks. Here we address the questions raised by the review as follows.

### 3.1 COMPARISON WITH CONTROLNET

In the paper we compare our method with ControlNet and also the depth conditioned Stable Diffusion v2.0 on the depth-to-RGB generation task qualitatively. To make the comparison more comprehensive, we additionally conduct quantitative comparison: we sample 30000 image-text pairs from

the COCO validation dataset, estimate depth of them by MiDaS, perform depth-to-RGB generation and compare the FID between the ground truth and the generated images. The results are shown in Tab. 1. The depth conditioned SD v2.0 (*SD v2.0d*) achieves the best performance among all, but the finetuning scale is much larger than other two methods. Our method shows comparable result with ControlNet, which is coincide with the qualitative observation in the paper. The marginal quantitative gap is negligible because of the randomness of generation. Also in the answer to reviewer *4SM3* (Sec. 4.1) we have an analysis of the relation between the FID score and the actual generation capability.

|      | SD v2.0d | ControlNet | JointNet (Ours) |
| ---- | -------- | ---------- | --------------- |
| FID  | 8.75     | 10.14      | 10.61           |

Table 1: Quantitative comparison of depth-to-RGB generation. SD v2.0d achieves the best score among all, with the cost of much heavier training. Our method is comparable with ControlNet.

## 3.2 ABLATION STUDY OF COMBINING CONTROLNET AND DIRECT EXTEND

We conduct the mentioned experiments and compare its joint generation quality with LDM3D, Direct Extend and JointNet via user study, whose details and results can be found in the answer to reviewer *4SM3* (Sec. 4.1). The satisfaction rate of *ControlNet + Direct Extend* and JointNet is 20.2% and 29.1% respectively, showing the effectiveness of our finetuning method.

The results is as expected because handling the output part is the key of finetuning a good joint generation model. As analyzed in the second paragraph of Sec. 3.1 of the main paper, directly extending the output will lead to catastrophic forgetting. We also show the training progression of this setup in Fig. 1 together with *Direct Extend* and JointNet. We can see the RGB generation quality is degraded similarly to *Direct Extend*. Also we found a concurrent work Wonder3D (Long et al., 2023), which uses cross-modality attention to keep the alignment of generated RGB and normal maps for text-to-3D generation, gives a similar analysis towards the *Direct Extend* strategy.

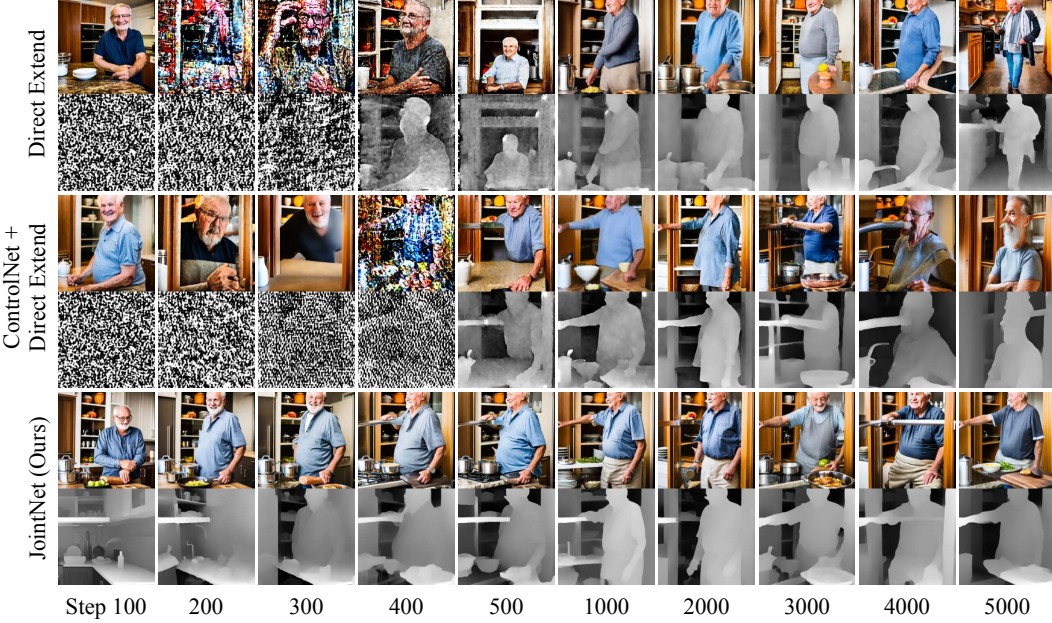

Figure 1: Progression of validation samples. The *Direct Extend* and the *ControlNet + Direct Extend* method lose the RGB generation capability in early steps, while JointNet successfully preserves it.

# 4 REVIEWER 4SM3

We thank the reviewer for the acknowledgement of the novelty of the network and finetuning design and the impressive qualitative quality. Here we address the questions raised by the review as follows.

## 4.1 QUANTITATIVE EVALUATION OF JOINT GENERATION

In the analysis of the quantitative evaluation in the main paper, we attribute the marginal gap to the fact that the global image descriptor used by FID cannot capture the difference of the details. We also find that these metrics cannot capture further improvement of image quality. For example, SDXL (Podell et al., 2023) is known to have much better generation quality, but achieves FID of 15.03 which is not significantly lower than other reported numbers. The authors of SDXL also extensively analyzed this issue in their technical report (Appendix Sec. F).

To this end, we conduct user study on overall quality of generated RGBD images. We compare JointNet with LDM3D, *Direct Extend*, and *ControlNet + Direct Extend*, which accepts joint modality by ControlNet and outputs as *Direct Extend*. We prepare 100 text prompts sampled from COCO validation set and let user choose the overall best generated RGBD image in terms of image-text consistency, image-depth consistency and the quality of image and depth themselves. The results are shown in Tab. 2. Among all the surveyed methods, JointNet achieves the highest satisfaction rate. We notice that LDM3D achieves the second highest score, which contradicts the qualitative and quantitative conclusion in the paper. After interviewing the participants, the main reason is that text-image consistency plays an important role in their decision making, which is LDM3D's strength, while the blurred image and the artifacts in the depth maps are not significant if the participants do not zoom in the samples.

| LDM3D | Direct Extend | ControlNet + Direct Extend | JointNet (Ours) |
|---|---|---|---|
| 27.0% | 23.7% | 20.2% | **29.1%** |

Table 2: User study of joint generation quality. For each of the 100 sampled text prompts, participants are told to select the best overall generated RGBD sample from 4 choices. Among all the surveyed methods, JointNet achieves the highest satisfaction rate. LDM3D also achieves good result due to its strong text-image consistency.

## 4.2 DISCUSSION OF THE IMAGE-TO-DEPTH EVALUATION AND THE NECESSITY OF JOINTNET

**Clarification of MiDaS+JointNet** Yes. The input image is resized to 384x384 and the corresponding depth map in the same size is estimated by MiDaS. Then we use JointNet to refine the depth conditioned on the image in the original size in a tile-based manner.

The *Ours (depth)* in Fig. 3 (main paper) refers to JointNet. But both image and depth are generated so it is not relevant to the image-to-depth task. The *JointNet (Ours)* in Fig. 4 (main paper) is from JointNet on its own. In Fig. 4 (main paper) we only select samples suitable for visualization. As shown in Fig. 2, there is a large amount of samples where only background area has ground truth but the background in the estimated depths are not distinguishable. And most large errors come from these kind of samples. For the samples shown in the main paper, the qualitative results are comparable. We will further analysis the quantitative results in the next paragraph.

**Analysis of the quantitative image-to-depth results** In Sec. 5.2 (main paper), we have discussed the quantitative results of the image-to-depth task. First of all, because the training data is prepared by MiDaS, we only expect the quality of diffusion depths to be comparable. One possible reason for the worse evaluation of JointNet is that MiDaS is directly trained on MegaDepth dataset while our model does not. Also, for the samples similar to what are shown in Fig. 2, it is difficult to align the estimation and ground truth, and thus the calculated error is random.

For the depth refinement task (*MiDaS+JointNet*), we can observe qualitative improvement in Fig. 5 (main paper). One possible reason for the marginal quantitative improvement is that object boundaries, where most improvements happen, are filtered out in the ground truth and thus not counted in the statistics.

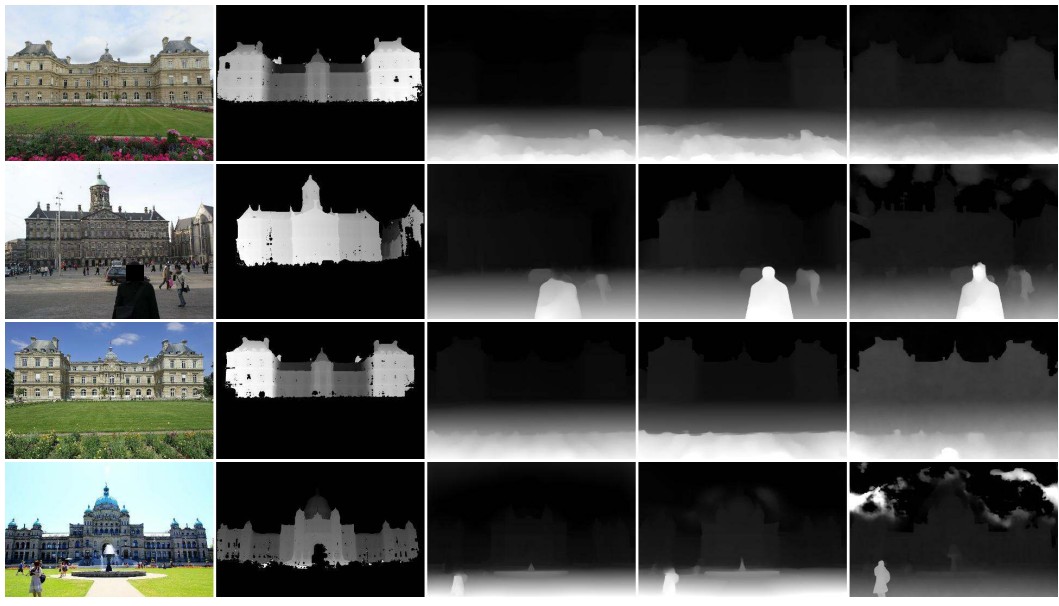

| Input Image | Ground Truth | MiDaS (BEiT-512) | MiDaS (DPT-Hybrid-384) | JointNet (Ours) |

Figure 2: Bad samples where ground truth and estimations are difficult to align. They are not suitable for visualization and generate large error. Recognizable human faces are masked.

**Necessity of JointNet**  If we only consider joint generation, Stable Diffusion followed by MiDaS can achieve the same task, and is already a widely used strategy. But because diffusion models essentially capture the distribution of the data, so it can do more than sample generation, e.g. SDEdit for image editing and tile-based generation for high resolution. Similarly, JointNet can additionally do depth/joint refinement and panorama generation compared with the two-step system. And it is not meaningless to be able to perform multiple tasks by a single model, which is the trend for very large deep learning models. Also, in a concurrent work Wonder3D (Long et al., 2023), which uses cross-modality attention to keep the alignment of generated RGB and normal maps for text-to-3D generation, the authors also discuss the differences between the two-step strategy (Sec. 5.6 and Fig. 7 in their paper) and show the benefits of joint generation.