# OpenReview forum: "JointNet: Extending Text-to-Image Diffusion for Dense Distribution Modeling"
_ICLR.cc/2024/Conference — ICLR 2024 poster_

### Official Review · Reviewer_4SM3 · 2023-10-30

**Soundness:** 3 good
**Presentation:** 3 good
**Contribution:** 2 fair
**Rating:** 5
**Confidence:** 3

**Summary:**

The paper introduces JointNet, a novel neural network architecture that seeks to model the joint distribution of images and a dense modality like depth maps. Originating from a pre-trained text-to-image diffusion model, the architecture generates a replica of the initial network for the dense modality, integrating it densely with the RGB branch. Simultaneously, it locks the RGB branch during fine-tuning, ensuring the preservation of the generalization capabilities of the pre-trained model. JointNet’s efficacy is validated across various applications, including joint RGBD generation, dense depth prediction, and 3D panorama generation, positioning the paper as a noteworthy contribution to the field of joint distribution modeling. It highlights a resourceful and efficient strategy that capitalizes on pre-trained models to enhance performance in computer vision tasks.

**Strengths:**

- The proposed methodology introduces a novel technique for modeling the joint distributions of images and dense labels.
- JointNet successfully maintains the intricate details in the depth maps, showcasing its refinement capabilities.
- The qualitative results are impressive, especially on the panoramic image and depth generation.

**Weaknesses:**

- The performance enhancement provided by the proposed network is marginal. According to Table 1, while JointNet outperforms Direct Extend in terms of IS, it falls short in FID and CLIP scores. Considering the network design is a primary contribution of this paper and differs significantly from Direct Extend, the results from Table 1 suggest only a slight improvement in performance due to the proposed network structure.
- Table 2 reveals that JointNet on its own yields poorer results compared to MiDaS. Although performance improves when combined with MiDaS, the enhancement is minimal, as seen in metrics such as AbsRel (improving from 0.0568 to 0.0561) and a decrease in RMSE (from 0.0601 to 0.0634). These results question the effectiveness of the proposed methods.
- Given the marginal performance difference between the proposed method and MiDaS, one could generate images using diffusion models, input them into MiDaS, and obtain high-quality images and depth maps, raising questions about the necessity of RGBD generation methods.

**Questions:**

- Does MiDaS+JointNet in Table 2 refer to initially extracting the depth map using MiDaS and subsequently refining it with JointNet? Could you please clarify?
- Are the results depicted by JointNet in Figure 4 and ‘Ours (depth)’ in Figure 3 associated with 'JointNet (Ours)' or 'MiDaS+JointNet' as mentioned in Table 2?
- If these results pertain to JointNet, the discrepancy between its quantitative performance in Table 2 (lower than MiDaS’s results) requires further explanation.

---

> ### Author Response · Authors · 2023-11-23
>
> \* We recommend reviewers to view the rebuttal in our global response, which is uploaded as the __supplementary material__. Figures and tables are also available.
>
> We thank the reviewer for the acknowledgement of the novelty of the network and finetuning design and the impressive qualitative quality. Here we address the questions raised by the review as follows.
>
> __Quantitative Evaluation of Joint Generation.__ In the analysis of the quantitative evaluation in the main paper, we attribute the marginal gap to the fact that the global image descriptor used by FID cannot capture the difference of the details. We also find that these metrics cannot capture further improvement of image quality. For example, SDXL is known to have much better generation quality, but achieves FID of 15.03 which is not significantly lower than other reported numbers. The authors of SDXL also extensively analyzed this issue in their technical report (Appendix Sec. F).
>
> To this end, we conduct user study on overall quality of generated RGBD images. We compare JointNet with LDM3D, _Direct Extend_, and _ControlNet + Direct Extend_, which accepts joint modality by ControlNet and outputs as _Direct Extend_. We prepare 100 text prompts sampled from COCO validation set and let user choose the overall best generated RGBD image in terms of image-text consistency, image-depth consistency and the quality of image and depth themselves. The results are shown in Tab. 2. Among all the surveyed methods, JointNet achieves the highest satisfaction rate. We notice that LDM3D achieves the second highest score, which contradicts the qualitative and quantitative conclusion in the paper. After interviewing the participants, the main reason is that text-image consistency plays an important role in their decision making, which is LDM3D's strength, while the blurred image and the artifacts in the depth maps are not significant if the participants do not zoom in the samples.
>
> __Clarification of MiDaS+JointNet.__ Yes. The input image is resized to 384x384 and the corresponding depth map in the same size is estimated by MiDaS. Then we use JointNet to refine the depth conditioned on the image in the original size in a tile-based manner.
>
> The _Ours (depth)_ in Fig. 3 (main paper) refers to JointNet. But both image and depth are generated so it is not relevant to the image-to-depth task. The _JointNet (Ours)_ in Fig. 4 (main paper) is from JointNet on its own. In Fig. 4 (main paper) we only select samples suitable for visualization. As shown in Fig. 2, there is a large amount of samples where only background area has ground truth but the background in the estimated depths are not distinguishable. And most large errors come from these kind of samples. For the samples shown in the main paper, the qualitative results are comparable. We will further analysis the quantitative results in the next paragraph.
>
> __Analysis of the quantitative image-to-depth results.__ In Sec. 5.2 (main paper), we have discussed the quantitative results of the image-to-depth task. First of all, because the training data is prepared by MiDaS, we only expect the quality of diffusion depths to be comparable. One possible reason for the worse evaluation of JointNet is that MiDaS is directly trained on MegaDepth dataset while our model does not. Also, for the samples similar to what are shown in Fig. 2, it is difficult to align the estimation and ground truth, and thus the calculated error is random.
>
> For the depth refinement task (_MiDaS+JointNet_), we can observe qualitative improvement in Fig. 5 (main paper). One possible reason for the marginal quantitative improvement is that object boundaries, where most improvements happen, are filtered out in the ground truth and thus not counted in the statistics.
>
> __Necessity of JointNet.__ If we only consider joint generation, Stable Diffusion followed by MiDaS can achieve the same task, and is already a widely used strategy. But because diffusion models essentially capture the distribution of the data, so it can do more than sample generation, e.g. SDEdit for image editing and tile-based generation for high resolution. Similarly, JointNet can additionally do depth/joint refinement and panorama generation compared with the two-step system. And it is not meaningless to be able to perform multiple tasks by a single model, which is the trend for very large deep learning models. Also, in a concurrent work Wonder3D, which uses cross-modality attention to keep the alignment of generated RGB and normal maps for text-to-3D generation, the authors also discuss the differences between the two-step strategy (Sec. 5.6 and Fig. 7 in their paper) and show the benefits of joint generation.

---

### Official Review · Reviewer_cBx7 · 2023-10-30

**Soundness:** 2 fair
**Presentation:** 3 good
**Contribution:** 2 fair
**Rating:** 6
**Confidence:** 3

**Summary:**

This paper proposed JointNet for modelling the joint distribution of images and dense information, e.g., depth and normal maps. JointNet is extended from a pre-trained text2img diffusion model by adding a copy of the original network as the new dense modality branch, densely connected with the RGB image branch, and while training the original network is locked. The proposed method extends the text2img diffusion models to more applications such as RGB-D generation, depth precautions, etc. Experiments show the improvements compared with previous methods.

**Strengths:**

This paper proposed a joint network that differs from previous fine-tuning methods or control-net which shows some novetly. The idea of using joint and RGB information bidirectional is natural and interesting. The instruction of the network architecture and training details are proposed. The proposed method could be used for other applications such as depth prediction.

**Weaknesses:**

Although the method compared with other methods on different applications to show the effectiveness. However, the experiments to show ablation study is not clear and also the comparison with control-net (the one that inspired the proposed method) is not clear.

**Questions:**

- How is the comparison between control net with the proposed method? Although the two methods focus on different targets and applications, it is still possible to test on the same task, e.g., change the noise from joint to regular RGB task, or evaluate on the inpainting task that works for both architecture.
- It is not clear if combine (a) and (b) in Figure 2 works well or not, e.g., use the same control net architecture but change the output as the joint and RGB, as based on Table 1 the numerical results are comparable and not that significant improved.
- The ablation study is not sufficient, it is good to show different strategies or detailed architectures.

---

> ### Author Response · Authors · 2023-11-23
>
> \* We recommend reviewers to view the rebuttal in our global response, which is uploaded as the __supplementary material__. Figures and tables are also available.
>
> We thank the reviewer for the acknowledgement of the novelty of the network and finetuning design, and the capability of handling various of downstream tasks. Here we address the questions raised by the review as follows.
>
> __Comparison with ControlNet.__ In the paper we compare our method with ControlNet and also the depth conditioned Stable Diffusion v2.0 on the depth-to-RGB generation task qualitatively. To make the comparison more comprehensive, we additionally conduct quantitative comparison: we sample 30000 image-text pairs from the COCO validation dataset, estimate depth of them by MiDaS, perform depth-to-RGB generation and compare the FID between the ground truth and the generated images. The results are shown in Tab. 1. The depth conditioned SD v2.0 (_SD v2.0d_) achieves the best performance among all, but the finetuning scale is much larger than other two methods. Our method shows comparable result with ControlNet, which is coincide with the qualitative observation in the paper. The marginal quantitative gap is negligible because of the randomness of generation. Also in the answer to reviewer _4SM3_ we have an analysis of the relation between the FID score and the actual generation capability.
>
> __Ablation Study of Combining ControlNet and Direct Extend.__ We conduct the mentioned experiments and compare its joint generation quality with LDM3D, Direct Extend and JointNet via user study, whose details and results can be found in the answer to reviewer _4SM3_. The satisfaction rate of _ControlNet + Direct Extend_ and JointNet is 20.2% and 29.1% respectively, showing the effectiveness of our finetuning method.
>
> The results is as expected because handling the output part is the key of finetuning a good joint generation model. As analyzed in the second paragraph of Sec.~3.1 of the main paper, directly extending the output will lead to catastrophic forgetting. We also show the training progression of this setup in Fig. 1 together with _Direct Extend_ and JointNet. We can see the RGB generation quality is degraded similarly to _Direct Extend_. Also we found a concurrent work Wonder3D, which uses cross-modality attention to keep the alignment of generated RGB and normal maps for text-to-3D generation, gives a similar analysis towards the _Direct Extend_ strategy.

---

### Official Review · Reviewer_UNzu · 2023-10-31

**Soundness:** 4 excellent
**Presentation:** 3 good
**Contribution:** 4 excellent
**Rating:** 8
**Confidence:** 3

**Summary:**

The authors introduce a neural network architecture for modeling the joint distribution of images and an additional "dense modality". The leading example of a dense modality is a depth map. Measurement of depth maps has important applications in designing self-driving cars, among many other use cases.

The procedure takes as an input a pre-trained text-to-image diffusion model. This model is copied and used as an initialization to model the additional dense modality. The weights of the original model are then locked and used to generate the RGB component. The copied model is then trained on a relevant dataset. In this way, the good performance of the input model for the RGB component is maintained.

**Strengths:**

The paper is very clearly written and provides a practical solution to an important problem. Their evaluation and assessment of their method is extensive and credible. They make several comparisons to alternative procedures, and provide a convincing argument that their approach is appropriate.

**Weaknesses:**

As the authors note, the procedure requires doubling the size of the input diffusion model. This structure becomes cumbersome if it is repeated for several modalities. I encourage future work on assessing the performance of a single additional model for the generation of several modalities.

**Questions:**

I have have no questions at this time.

---

> ### Author Response · Authors · 2023-11-23
>
> \* We recommend reviewers to view the rebuttal in our global response, which is uploaded as the __supplementary material__. Figures and tables are also available.
>
> We thank the reviewer for the appreciation of the contribution and clear paper writing. Here we address the questions raised by the review as follows.
>
> __Additional Computational Complexity for Joint Modalities.__ Thanks for the suggestion on the possible future improvement and we agree with it. Because the additional branch needs to be trained to adapt to a new domain after all, preserving the original knowledge is less important than the RGB branch, and thus we can apply the direct extend strategy to merge multiple modalities into the joint branch. One disadvantage is the reduced flexibility. If each modality has its own JointNet, users can pick only the needed ones and use them in a plug-and-play manner just like multi-ControlNet.

---

### Official Review · Reviewer_yguJ · 2023-11-06

**Soundness:** 2 fair
**Presentation:** 2 fair
**Contribution:** 2 fair
**Rating:** 3
**Confidence:** 4

**Summary:**

The method tries to use text to image diffusion models to address different applications in vision such as monocular depth estimation.

**Strengths:**

+ The work seems to be well-motivated and has addressed an important field.

**Weaknesses:**

- The work is not evaluated that well. For example, monocular depth estimation is not compared with several SOTA methods.

- The work is inferior compared to the other methods.

- The work tries to do many things but is irrelevant in all of them.

- The work needs to be presented well with proper comparisons and illustrations.

**Questions:**

What is the primary contribution of this work?

What benefit it has for the user provided that this method is not giving best results for all the applications mentioned?

---

> ### Author Response · Authors · 2023-11-23
>
> \* We recommend reviewers to view the rebuttal in our global response, which is uploaded as the __supplementary material__. Figures and tables are also available.
>
> We thank the reviewer for the acknowledgement of the motivation of this work and the importance of the task. Here we address the questions raised by the review as follows.
>
> __Clarification on the Contribution.__ We would like to clarify that the core idea is using joint distribution modeling to construct a large vision model capable of various of downstream tasks, instead of pushing the SOTA of each single downstream task. Because training large models is more and more expensive, it is desirable to design a general training objective for a multi-tasking model. For the performance gap of image-to-depth task, because our depth training data is prepared by the mentioned SOTA depth estimation network and we do not directly train on (the training split of) the depth datasets the benchmark uses, our model may not be able to outperform the SOTA depth estimator, which is indeed a limitation. Possible improvements include adding the depth datasets for training and finetuning the VAE for the joint modalities. More explanation can be found in the answer to reviewer _4SM3_.
>
> On the other hand, the joint diffusion model unlocks several new usage such as tile-based consistency handling for high-res RGBD generation (e.g., RGBD panorama), which cannot be achieved by the traditional discriminative model of monocular depth estimation. To summarize, we believe our work demonstrates the potential of the joint training scheme, and there are possible ways of improvement to further push the performance on each downstream task.

---

### Author Response · Authors · 2023-11-21

We thank all the reviewers for their insightful comments. We have prepared a global response and uploaded as the supplemental material (the original supp. material is attached in the main paper).

---

### Meta-Review · Area_Chair_VABV · 2023-12-10

**Metareview:**

This paper tries to model the image and the dense depth map jointly from a pre-trained text-to-image diffusion model.

The reviewers recognized the strengths in terms of the novelty of the idea and good qualitative results.
Meanwhile, there were concerns about
* the marginal quantitative gains compared with the baseline, Direct Extend
* lack of comparisons with SoTA method, ControlNet
* depth accuracy falling behind a single-image depth prediction method, MiDaS
* insufficient ablation study.

In the rebuttal, the authors
* admitted the marginal gains issue, however, claimed justification for the proposed method due to the extra capability brought from the joint modeling of image and depth. In addition, the authors performed a user study result to provide further comparisons.
* claimed the refinement using both MiDaS and the proposed JointNet brings improvement.
* provided an ablation study in the rebuttal.

As the authors addressed the concerns, I would recommend acceptance of this paper. The authors are recommended to best reflect the updates in the rebuttal to the main manuscript for clarity. Especially, please provide a sufficient discussion on the quantitative gains issue by different metrics.

**Justification For Why Not Higher Score:**

While the reviewer response addressed the concerns, the marginal gains issue is still valid. The authors are suggested to provide further analysis on the final manuscript.

**Justification For Why Not Lower Score:**

The concerns from the reviewers are addressed in the rebuttal process.

---

### Decision · Program_Chairs · 2024-01-16

Accept (poster)